# A Smooth Non-Iterative Local Polynomial (SNILP) Model of Image Vignetting

**DOI:** 10.3390/s21217086

**Published:** 2021-10-26

**Authors:** Artur Bal, Henryk Palus

**Affiliations:** 1Department of Data Science and Engineering, Silesian University of Technology, Akademicka 16, 44-100 Gliwice, Poland; 2Biotechnology Center, Silesian University of Technology, Bolesława Krzywoustego 8, 44-100 Gliwice, Poland

**Keywords:** image vignetting, lens shading, vignetting correction, flat-field correction, vignetting modeling, approximation function, low-level vision, embedded vision systems

## Abstract

Image vignetting is one of the major radiometric errors, which occurs in lens-camera systems. In many applications, vignetting is an undesirable phenomenon; therefore, when it is impossible to fully prevent its occurrence, it is necessary to use computational methods for its correction in the acquired image. In the most frequently used approach to the vignetting correction, i.e., the flat-field correction, the usage of appropriate vignetting models plays a crucial role. In the article, the new model of vignetting, i.e., Smooth Non-Iterative Local Polynomial (SNILP) model, is proposed. The SNILP model was compared with the models known from the literature, e.g., the polynomial 2D and radial polynomial models, in a series of numerical tests and in the real-data experiment. The obtained results prove that the SNILP model usually gives better vignetting correction results than the other aforementioned tested models. For images larger than UXGA format (1600×1200), the proposed model is also faster than other tested models. Moreover, among the tested models, the SNILP model requires the least hardware resources for its application. This means that the SNILP model is suitable for its usage in devices with limited computing power.

## 1. Introduction

### 1.1. Problem of Vignetting

The image vignetting is a phenomenon of the reduction of the brightness of an image from its optical center toward its edges. The vignetting is often an unintended and undesired effect, and its characteristics depends on optical properties of the lens-camera system. Considering the causes of vignetting, there are four main types of this phenomenon [1,2], listed it in the order of the place of its occurrence from an imaged scene to an image sensor, i.e., mechanical vignetting, optical vignetting, natural vignetting, and pixel vignetting. Mechanical vignetting refers to the occlusion of the light path by elements of lens-camera system, such as a filter mounted on a lens. This type of vignetting usually causes a 100% reduction in image brightness, which means a complete loss of information. Therefore, this type of vignetting will not be taken into consideration in this article. The next type of vignetting is optical vignetting, which refers to the light fall-off caused by the blockage of off-axis incident light inside the lens body. The amount of blocked light depends on the physical dimensions of the lens, particularly the lens focal length, diameter of lenses in the lens body, and the lens aperture setting. Natural vignetting refers to the light fall-off related to the geometry of the image-forming system. Pixel vignetting refers to the light fall-off related to the angle-dependent sensitivity of the image sensor pixels.

The effect of vignetting is undesirable in many applications, especially when there is a need for stitching images for creating panoramic images [3,4] or mosaic images [5,6], radiometric, or quantitative analysis of images [7,8], e.g., in such areas as microscopy [6,9,10], micro CT imaging [11], and remote sensing [12,13]. The best way to reduce vignetting is to remove its causes, e.g., by the usage of a lens with appropriate characteristics or by careful choice of exposure parameters. However, the usage of such solutions is not always possible nor brings the desired result. In such cases, the only solution is to use a software-based vignetting correction method.

### 1.2. Vignetting Correction

Probably the most frequently used approach to vignetting correction is based on the idea of ”flat field correction.” In this approach, the vignetting is estimated based on reference image of vignetting $I_{V}$, which presents uniformly illuminated flat surface with a uniform color. The only source of brightness differences across $I_{V}$ is the vignetting of the used lens-camera system. The way an image $I_{V}$ is formed can be described by the following formula:(1)$$I_{V}\left(i,j\right)=V\left(i,j\right)\;\cdot \;I_{flat}\left(i,j\right),$$
where *V* is a real vignetting of the analyzed lens-camera system, $I_{flat}$ is an ideal image of a scene with reference flat surface, and (i,j) are, respectively, horizontal and vertical pixel coordinates. Vignetting estimator $\tilde{V}$ is established during approximation process using assumed vignetting model *VM* and image $I_{V}$, i.e.,
(2)$$\tilde{V}\left(i,j\right)=\mathrm{approx}\left(VM,I_{V}\left(i,j\right)\right),$$
where $\tilde{V}\in \left(0,1\right]$. For obtaining image $\tilde{I}$ with corrected vignetting, which is based on the acquired image *I*, the following calculations must be done:(3)$$\tilde{I}\left(i,j\right)=I\left(i,j\right)\;\cdot \;\tilde{V}{\left(i,j\right)}^{-1}.$$

For the best correction results the images $I_{V}$ and *I* should be acquired using the same lens-camera system, its parameters (e.g., focal length for zoom lenses), and it exposure parameters (e.g., aperture stop number of the lens). In practice, for real-time vignetting correction, (Equation 3) is realized using lookup table.

The flat field correction is, of course, not the only approach which realizes the vignetting correction. The others are based on, e.g., physically-based models of vignetting [14,15] and analysis of a sequence of images [1,2,3,16] or a single image [17,18,19,20] of a natural scene to estimate the vignetting. However, the usage of these approaches is connected with many limitations. In the case of methods, which use the physically-based approach, detailed information about the parameters of a lens-camera system is needed, which is often unavailable. Additionally, the obtained vignetting model is usually limited to only one type of vignetting. In the case of both aforementioned image-based approaches, the vignetting estimation is usually obtained as a result of finding an optimal value an objective function for an assumed radial vignetting model; however, this assumption limits the number of lens-camera systems for which these approaches can be used. Moreover, the effectiveness of image-based methods depends strongly on the precision of localization of corresponding pixels in images sequence, which requires usage of an additional correction of lens distortion. It should also be mentioned that the effectiveness of these methods strongly depends on scene uniformity.

In comparison to the aforementioned approaches, the flat-field-based approach is more universal; this method can be used for virtually any lens-camera system. The adjustment of the flat-field method to the analyzed system is the matter of the usage of the vignetting model, which is appropriate for the actual vignetting occurring in the analyzed system. This mainly means the usage of vignetting model, for which assumptions of its use are consistent with the features of vignetting of the analyzed lens-camera system.

Over the years, in the literature, different vignetting models based on the idea of parametric approximation have been presented, e.g., polynomial model [1,21,22], exponential polynomial model [21], hyperbolic cosine model [23], Gaussian function [24], and radial polynomial model [25]. The last three aforementioned vignetting models belong to a popular approach, which assumes that the actual vignetting *V* of lens-camera system has a cylindrical symmetry. This means that vignetting can be modeled using radial function, i.e., the lens-camera vignetting V(i,j) is a function of *r*, where *r* is a distance between pixel p(i,j) and optical center *C* (with coordinates $\left(i_{C},j_{C}\right)$) of the lens-camera system. The use of the radial vignetting assumption simplifies the process of searching for the vignetting estimator; the approximation of the 2D function is replaced by the approximation of the 1D function. On the other hand, searching for optical center is not a trivial task [26]; its realization involves additional calculations and is a source of potential errors. Moreover, the usage of a radial function limits the applicability of this method.

The choosing of the proper vignetting model is not the only problem related to flat-field correction. The other problem is the preparation of a reference scene, especially the uniform illumination of the reference surface is a problem. It should be emphasized that the quality of the obtained $\tilde{V}$ estimator strongly depends on the luminance uniformity of the reference scene.

### 1.3. Objective of the Work

The assumption about radial vignetting can be used for most lens-camera systems. However, there is a large group of systems for which this assumption is not applicable, e.g., industrial lenses designed with reducing vignetting in mind, perspective-control and focal plane-control lenses (shift and tilt lenses), and anamorphic lenses. Therefore, there is a need for an universal vignetting model which can be used for different lens-camera systems.

Good examples of such an universal solution are the vignetting models proposed by Sawchuk in 1977, in Reference [21], in the form of polynomial 2D (P2D) and exponential polynomial 2D vignetting models. In both models, vignetting estimators are found as a result of seeking a solution to the multiple regression problem. Another universal solution was proposed in 2004 by Yu [23]. The proposed method is based on the idea of wavelet denoising and decimation. The common feature of the aforementioned methods is the relatively high computational complexity of these methods. Due to this feature, these methods are not very suitable for implementation in devices with relatively low computing power or memory resources, such as, e.g., portable devices, smart cameras, vision sensors, embedded vision systems, and unmanned vehicles. Hence, there is a need to develop a universal vignetting model that will combine relatively low computational and memory complexity with the high accuracy of vignetting correction.

A possible solution to this problem uses the idea of decomposition of the 2D approximation problem into many interrelated 1D approximation problems. This approach was presented by Reference [27,28,29]. The results obtained so far in this area are briefly described and summarized in Section 2. The aim of the conducted research was to improve the vignetting model based on this idea. The new vignetting model (SNILP) that meets this requirement is presented in Section 3. Section 4 presents results of experiment that was performed to compare the quality of the vignetting correction results obtained from the new model with the results obtained from selected vignetting models described in the literature. In Section 5, comparisons of others features of the compared models are presented. Lastly, Section 6 contains a summary of the research.

## 2. Vignetting Models Based on Decomposition

### 2.1. Local Parabolic Model

The problem of vignetting estimation is, in fact, the problem of seeking an approximation function of 2D data. As presented in Reference [27], in the form of *Local Parabolic* model, the 2D multiple approximation problem for image of size M×N pixels can be decomposed into M+N simple linear regression problems using the following formula:(4)$$\tilde{V}\left(i,j\right)=\frac{1}{2}\left({\tilde{V}}_{x_{j}}\left(i\right)+{\tilde{V}}_{y_{i}}\left(j\right)\right),$$
where
(5)$$\begin{array}{ccc} {\tilde{V}}_{x_{j}}\left(i\right) & = & a_{x_{j},2}\;i^{2}+a_{x_{j},1}\;i+a_{x_{j},0},\\ {\tilde{V}}_{y_{i}}\left(j\right) & = & a_{y_{i},2}\;j^{2}+a_{y_{i},1}\;j+a_{y_{i},0}, \end{array}$$
are, respectively, values of approximation functions of degree 2 obtained for the *i*-th point of the *j*-th horizontal line and the *j*-th point of the *i*-th vertical line of the input image. The parameters *a* of functions (Equation 5) are obtained as a result of repeated 1D polynomial regression of degree 2, which is performed along each horizontal and each vertical line of the input image.

### 2.2. Local Polynomial (LP) Model

The more general *Local Polynomial* (LP) model of vignetting was proposed in Reference [28]. In this model, the degree of polynomial *s* used as 1D regression function is not limited to degree of 2 but can be freely chosen; hence, ${\tilde{V}}_{x_{j}}$ and ${\tilde{V}}_{y_{i}}$ are calculated as follows:(6)$$\begin{array}{ccc} {\tilde{V}}_{x_{j}}\left(i\right) & = & a_{x_{j},s}\;i^{s}+\ldots +a_{x_{j},1}\;i+a_{x_{j},0},\\ {\tilde{V}}_{y_{i}}\left(j\right) & = & a_{y_{i},s}\;j^{s}+\ldots +a_{y_{i},1}\;j+a_{y_{i},0}, \end{array}$$
where *s* is the assumed degree of regression polynomials.

The LP model gives more exact approximation results of vignetting than Local Parabolic model. However, the more exact analysis of the LP model shows that the obtained surface of vignetting estimator is non-smooth and ragged.

### 2.3. Smooth Local Polynomial (SLP) Model

As a solution to this problem, in Reference [29], the *Smooth Local Polynomial* (SLP) model was proposed. The core idea of this method is the iterative repetition of the LP model estimation, i.e.,
(7)$${\tilde{V}}_{k}\left(i,j\right)=\mathrm{LP}\left({\tilde{V}}_{k-1}\left(i,j\right)\right),$$
where ${\tilde{V}}_{k}$ is a result of the *k*-th iteration of the SLP, and, for k=0, it holds that
(8)$${\tilde{V}}_{0}\left(i,j\right)=I_{V}\left(i,j\right),$$
which is the input image for the SLP model. It follows directly from the definition of the SLP model that, for k=1, the results obtained from this model are the same as the results obtained from the LP model.

### 2.4. Essential Properties of the LP and SLP Models

The following belong to the essential properties of each of the vignetting models: accuracy of reproducing the actual vignetting *V* on the basis of the obtained image or images $I_{V}$, features of the obtained model (such as, e.g., surface smoothness), and ease of use of the model. Due to the fact that, when determining the vignetting estimate for real lens-camera systems, it is difficult to reliably verify the correctness of the obtained model, it was decided to perform numerical simulations.

The same test conditions were used in all simulations presented in this section, as well as in Section 3.2. The simulation conditions are described in detail in Appendix A. For comparison of vignetting models, lenses with three focal lengths were simulated for 135 film format, i.e., f= 24 mm, 50 mm, and 300 mm. This selection of focal lengths includes the most important focal lengths of lenses (converted to the appropriate image formats used in a given application), which are used for, e.g., photography, film-making, monitoring, machine vision, and remote sensing.

The use of vignetting models based on the idea of image decomposition requires the determination of the degree of the polynomial that will be used to approximate individual lines of the image. In the case of the presented simulation results, the following polynomial degree values were used, respectively, to the aforementioned focal lengths: 10, 6, and 2. Such a selection of the degrees of approximation polynomials results from the fact that, for these degrees of the polynomial, the best results (in terms of Root Mean Square Error) of the vignetting estimation for the tested focal lengths were obtained. To obtain statistically reliable results, the simulations were repeated for each lens 100 times.

The most important property of vignetting model is probably its accuracy. Thanks to the simulation, the exact shape of the *V* vignetting is known. Hence, for evaluation of model accuracy, the commonly used difference measures, such as Mean Absolute Error (MAE)
(9)$$\mathrm{MAE}\left(\tilde{V},V\right)=\frac{1}{MN}\sum_{i=1}^{M}\sum_{j=1}^{N}|\tilde{V}\left(i,j\right)-V\left(i,j\right),|$$
or Root-Mean-Square Error (RMSE)
(10)$$\mathrm{RMSE}\left(\tilde{V},V\right)=\sqrt{\frac{1}{MN}\sum_{i=1}^{M}\sum_{j=1}^{N}{\left(\tilde{V}\left(i,j\right)-V\left(i,j\right)\right)}^{2}},$$
can be used. Figure 1 presents values of those measures obtained for the simulated data.

Based on the knowledge about the causes of vignetting, it can be concluded that the vignetting function should have a smooth surface. As previously mentioned, surfaces obtained from the LP model are non-smooth; thus, they differ from the vignetting of real lens-camera systems. Figure 2 presents a comparison of smoothness of exemplary results obtained from the LP model and different numbers of iterations of the SLP model. The presented results confirm the significant improvement in smoothness obtained by using the SLP model compared to the LP model.

The smoothness of the surface of vignetting estimator $\tilde{V}$ can be evaluated not only visually but also using objective measures, such as *Mean Local Standard Deviation* (MLSD) and *Mean Local Slope Corrected Deviation* (MLSCD). The usage of MLSD measure for smoothness evaluation is based on the idea of Local Standard Deviation (LSD), which was used for decades by image processing community, in original [30,31] or slightly modified form, for evaluation of local changes in images. LSD is calculated in moving window $w_{ij}$ of size $w_{x}\times w_{y}$, where coordinates (i,j) defined the position of the window center in the analyzed image *I* of resolution M×N, i.e.,
(11)$$w_{ij}=I\left(i-r_{x}:i+r_{x},j-r_{y}:j+r_{y}\right),$$
where $2r_{x}+1=w_{x}$, and $2r_{y}+1=w_{y}$; notation a:b denotes the range of values from *a* to *b* with increment +1. In this work, MLSD is calculated as follows:(12)$$\mathrm{MLSD}\;\left(I\right)=\frac{1}{\left(M-w_{x}+1\right)\left(N-w_{y}+1\right)}\sum_{i=r_{x}+1}^{M-r_{x}}\sum_{j=r_{y}+1}^{N-r_{y}}\mathrm{LSD}\left(w_{ij}\right),$$
where
(13)$$\mathrm{LSD}\left(w_{ij}\right)=\sigma \left(w_{ij}\right)=\sqrt{\frac{1}{w_{x}\;w_{y}}\sum_{g=1}^{w_{x}}\sum_{h=1}^{w_{y}}{\left(w_{ij}\left(g,h\right)-{\bar{w}}_{ij}\right)}^{2}},$$
and
(14)$${\bar{w}}_{ij}=\frac{1}{w_{x}\;w_{y}}\sum_{g=1}^{w_{x}}\sum_{h=1}^{w_{y}}w_{ij}\left(g,h\right).$$

The idea of MLSCD is based on the idea of MLSD with one important modification; in the case of MLSCD locally (i.e., inside of moving window $w_{ij}$), the standard deviation $\sigma (w_{ij})$ of pixel values is not calculated, such as in LSD, in relation to the average value ${\bar{w}}_{ij}$ of pixels inside $w_{ij}$, but in relation to a local 2D polynomial approximation ${\mathrm{PolyApp}}_{2D}^{s}\left(w_{ij}\right)$ of degree *s* of image *I* surface covered by $w_{ij}$. This change results in a much lower sensitivity of the MLSCD measure to different slopes of analyzed surfaces; thanks to this, the proposed measure suits better to measure the surface smoothness than the MLSD measure. This property of MLSD results from the fact that values of LSD are strongly influenced by the local slope of the analyzed surfaces; therefore, in the LSD measure, the fluctuations of surface smoothness are dominated by changes of pixel values caused by a surface slope. The MLSCD measure is calculated as follows:(15)$$\mathrm{MLSCD}\;\left(I\right)=\frac{1}{\left(M-w_{x}+1\right)\left(N-w_{y}+1\right)}\sum_{i=r_{x}+1}^{M-r_{x}}\sum_{j=r_{y}+1}^{N-r_{y}}\sigma \left(\Delta_{ij}\right),$$
where
(16)$$\sigma \left(\Delta_{ij}\right)=\sqrt{\frac{1}{w_{x}\;w_{y}}\sum_{g=1}^{w_{x}}\sum_{h=1}^{w_{y}}{\left(\Delta_{ij}\left(g,h\right)-{\bar{\Delta }}_{ij}\right)}^{2}},$$
and
(17)$$\Delta_{ij}=w_{ij}-{\mathrm{PolyApp}}_{2D}^{s}\left(w_{ij}\right),$$
(18)$${\bar{\Delta }}_{ij}=\frac{1}{w_{x}\;w_{y}}\sum_{g=1}^{w_{x}}\sum_{h=1}^{w_{y}}\Delta_{ij}\left(g,h\right).$$

It is worth noting that, using s=0 in (Equation 17), the MLSCD measures come down to calculating the value of MLSD. In the work, the windows size $w_{x}=w_{y}=7$ were used for both MLSD and MLSCD measures; in the case of MLSCD, s=2 was used. To reduce the influence of the border effect associated with the use of a moving window on the smoothness evaluation results, the range of coordinates of moving window centers used for calculation of MLSD and MLSCD (Equations (Equation 12) and (Equation 15)) is smaller than the actual size of the analyzed images.

Figure 3 presents the comparison of MLSD and MLSCD values obtained for LP and SLP (for k=1,…,30, where, as mentioned earlier, for k=1, it holds that $LP\left(I_{V}\right)=SLP\left(I_{V}\right)$) models for simulated vignetting of three focal length *f* of simulated lenses (see Appendix A for experiment details). The presented results show that the SLP model outperforms the LP model at the price of *k* times larger computational complexity. At this moment, the practical question that arises is as follows: Is it possible to obtain estimation results with quality similar to those obtained from the SLP model for large *k*, with the computational cost close to the cost required for the LP model?

## 3. SNILP—A New Vignetting Model

### 3.1. Model Description

To simplify further considerations, the whole process of repeated 1D polynomial regression is based on the horizontal and the vertical lines, and then calculation, based on these results, approximated values of the input image *I* for point (i,j), which, from now on, is, respectively, denoted as ${\mathrm{PolyApp}}_{x}^{s}\left(I,(i,j)\right)$ and ${\mathrm{PolyApp}}_{y}^{s}\left(I,(i,j)\right)$; *s* is a degree of the used 1D polynomial regression and, later, 1D approximation function.

In the SLP model, the approximations of the input image $I_{V}$ based on independently calculated approximations for horizontal and vertical lines, i.e., ${\tilde{V}}_{x}$ and ${\tilde{V}}_{y}$. The values of these estimators depend only on the input image and the chosen degree *s* of regression polynomial. The usage of the opposite approach is the foundation idea used in the new *Smooth Non-Iterative Local Polynomial Vignetting* (SNILP) model. The values of approximations ${\mathrm{PolyApp}}_{x}^{s}$ and ${\mathrm{PolyApp}}_{y}^{s}$ of $I_{V}$ are calculated in sequences, e.g., as a first approximation in horizontal direction is calculated, and, then, based on these results, the approximation in vertical direction is calculated. This procedure can be written formally as
(19)$${\tilde{V}}_{x}\left(i,j\right)={\mathrm{PolyApp}}_{x}^{s}\left(I_{V},\left(i,j\right)\right),$$
(20)$$\tilde{V}\left(i,j\right)={\mathrm{PolyApp}}_{y}^{s}\left({\tilde{V}}_{x},\left(i,j\right)\right).$$

All operations used for calculation needed in the SNILP model are linear. Therefore, the sequence of the directions in which the regression is performed does not matter; hence,
(21)$$\tilde{V}\left(i,j\right)={\mathrm{PolyApp}}_{y}^{s}\;\left({\mathrm{PolyApp}}_{x}^{s}\;\left(I_{V},\;\left(\forall_{i},\forall_{j}\right)\right),\;\left(i,j\right)\right)\;=\;{\mathrm{PolyApp}}_{x}^{s}\;\left({\mathrm{PolyApp}}_{y}^{s}\;\left(I_{V},\;\left(\forall_{i},\forall_{j}\right)\right),\left(i,j\right)\right).$$

Since, in practice, the acquisition parameters of image $I_{V}$ are selected so that there is no saturation phenomenon in it, therefore, the following condition is fulfilled $\max\;(\tilde{V})<1$. Hence, it is necessary to normalize the range of $\tilde{V}$ to the range (0,1], which is performed as follows:(22)$$\tilde{V}\left(i,j\right):=\frac{\tilde{V}\left(i,j\right)}{\max_{i,j}\;\left(\tilde{V}\left(i,j\right)\right)}.$$

From now on, $\tilde{V}$ denotes the vignetting estimate in the normalized range.

### 3.2. Selected Properties of the SNILP Model

Analyzing the properties of the SNILP model, it is worth starting with paying attention to the fact that, in practice, the relationship (Equation 21) is satisfied with a certain error. The occurrence of this numerical error is related to the fact that the change of the order of directions in which the 1D approximations are calculated (e.g., first along *X* axis, and then along *Y* axis) may cause some differences in estimation results. However, on the basis of the carried out tests, the results of which are presented in Table 1 and Figure 4, it can be concluded that, for each repetition, the maximal error that arises due to a change in the approximations directions order was not greater than $2\times 10^{-12}$; so, they are irrelevant, even in the case of use 16-bit analog-to-digital converters in a camera image processor. The conditions of numerical test were identical to those used in Section 2.4.

Idempotence is the property of some mathematical operations or algorithms, whereby its multiple applications for the same data and the same parameters does not affect the obtained result. In the case of the SNILP model, it means
(23)$$\tilde{V}\left(i,j\right)=\mathrm{SNILP}\;\left(I_{V}\left(i,j\right)\right)={\mathrm{SNILP}}_{k}\;\left({\mathrm{SNILP}}_{k-1}\;\left(\ldots {\mathrm{SNILP}}_{2}\;\left(\mathrm{SNILP}\;\left(I_{V}\left(i,j\right)\right)\right)\ldots \right)\right),$$
where *k* is a number of repetition usage of the SNILP model. In practice, however, because of the presence of the numerical error in polynomial approximation procedures,
(24)$$\mathrm{SNILP}\;\left(I_{V}\left(i,j\right)\right)={\mathrm{SNILP}}_{k}\;\left({\mathrm{SNILP}}_{k-1}\;\left(\ldots {\mathrm{SNILP}}_{2}\;\left(\mathrm{SNILP}\;\left(I_{V}\left(i,j\right)\right)\right)\ldots \right)\right)+\varepsilon \left(i,j\right),$$
where ε(i,j) is the error value for point (i,j) related to numerical errors.

Figure 5 presents mean errors $\bar{\varepsilon }\left(i,j\right)$ calculated for different numbers *k* of repeated usage of the SNILP model, the values $\bar{\varepsilon }\left(i,j\right)$ calculated on the basis of 100 repetitions of the experiment. One additional usage (k=2) of the SNILP model changes, on average, the results for each point (i,j) not more than $10^{-13}$, and such changes do not have practical meaning. The influence of numerical errors is, of course, more visible for larger *k*, but, even for k=25, the values of $\bar{\varepsilon }\left(i,j\right)$ are not greater than $10^{-11}$. The maximum difference between RMSE values caused by repetitive usage of the SNILP model does not exceed $5\times 10^{-13}$. These results show that usage of the SNILP model is an idempotent operation.

### 3.3. Comparison of the SLP and SNILP Models

Figure 1 presents a comparison of MAE and RMSE values for SLP (box-plot) and SNILP (orange lines—solid line is a median) models. The presented plots show that differences between results, obtained from the SLP and SNILP models, gradually decrease with the execution of subsequent iterations of the SLP model. Based on these results, the following supposition can be formulated:(25)$$\lim_{k\to \infty }{\mathrm{SLP}}_{k}^{s}\left(I_{V}\right)={\mathrm{SNILP}}^{s}\left(I_{V}\right),$$
i.e., vignetting estimates obtained from the SNILP model are, for the same degree *s* of approximation polynomials, asymptotes for a series of estimates obtained from the successive iterations of the SLP model. This supposition is confirmed by the results (Figure 3) of measure of smoothness, using the MLSD and MLSCD measures, of vignetting estimate surfaces obtained from both models.

Comparing the SLP model with the SNILP model, it should be recalled that the result for the SNILP model is obtained in one step, when a qualitative similar result from the SLP model requires *k* iterations. In the SLP model, for the image of size M×N pixels, one iteration step includes *N* approximations in the horizontal direction, *M* approximations in the vertical direction, and MN summations and MN splits for calculation of mean from these approximations. In the case of the SNILP model, a whole calculation requires only *N* approximations in the horizontal direction, as well as *M* approximations in the vertical direction. Hence, assuming a similar quality of the results obtained from the SLP and SNILP models and the same polynomial degree *s* used in both models, the following relationship between the computational complexities of these models is true:(26)$$O\left(\mathrm{SNILP}(M,N)\right)\;<\;\frac{1}{k}O\left(\mathrm{SLP}(k,M,N)\right).$$

## 4. Experimental Results

### 4.1. Assumptions and Conditions of the Experiment

The aim of conducted experiment was a comparison, based on real data, of the selected, known from the literature, vignetting models with the SNILP model. The quality of vignetting model can be evaluated by the analysis of the result image:(27)$${\tilde{I}}_{flat}=I_{V}\;\cdot \;{\tilde{V}}^{-1},$$
i.e., results of correction of the vignetting image $I_{V}$, with the usage of vignetting estimates $\tilde{V}$ obtained using different vignetting models. Ideal correction results should be flat, i.e., all pixels in the corrected image ${\tilde{I}}_{flat}$ should have the same value. Of course, in the case of real data, this ideal result cannot be achieved because of the presence of noise in the input $I_{V}$ image. In such cases, the pixels in the ${\tilde{I}}_{flat}$ image should have similar values with the possible minimal dispersion. Good measures of dispersion are standard deviation (STD) and interquartile range (IQR); therefore, these measures were used for quantitative assessment of the vignetting correction results. Hence, a lower value of $\mathrm{STD}\left({\tilde{I}}_{flat}\right)$ or $\mathrm{IQR}\left({\tilde{I}}_{flat}\right)$ means better ability of the analyzed vignetting model to adapt to the real vignetting.

Since the $I_{V}$ image contains noise, and the estimation result $\tilde{V}$ may differ significantly from the vignetting of a given camera, it is possible that the ${\tilde{I}}_{flat}$ image will contain pixels with values beyond the range of [0,255], i.e., the range of pixel values for cameras used in the experiment. Therefore, the correction result, i.e., the image ${\tilde{I}}_{flat}$, before its evaluation, is subjected to the operation of truncation of pixel values according to the formula
(28)$${\tilde{I}}_{flat}\left(i,j\right):=\left\{\begin{array}{ccc} 0 & \mathrm{for} & {\tilde{I}}_{flat}\left(i,j\right)<0\\ {\tilde{I}}_{flat}\left(i,j\right) & \mathrm{for} & {\tilde{I}}_{flat}\left(i,j\right)\in \left[0,255\right]\\ 255 & \mathrm{for} & {\tilde{I}}_{flat}\left(i,j\right)>255 \end{array}.\right.$$

The data analyzed in the experiment were acquired using three different webcams, namely: Logitech C920, Xiaomi IMILAB CMSXJ22A, and A4Tech PK-910H (Figure 6), which, in the article, are denoted, respectively, as Cam-A, Cam-B, and Cam-C. In the case of webcams, their manufacturers do not provide detailed technical data, such as, e.g., lens focal length, lens speed, etc. However, the webcams used in the experiment were carefully selected to differ in their parameters, particularly the field of view and geometric distortions; comparison of the webcams main technical data is presented in Table 2. Such a selection of cameras allows for carrying out the experiment for possibly different examples of real vignetting.

As a flat-field surface needed for acquiring vignetting image ($I_{V}$), the uniformly back-lighted milky poly(methyl methacrylate) (PMMP, ”plexiglass”) panel of size 50 cm×100 cm and thickness 2.5 mm was used. As a light source, the Asus PA248QV graphic monitor displaying a white screen with brightness set to maximum, i.e.,  300 lx, was used. The plexiglass panel was carefully positioned parallel to the monitor screen surface. In order to position the camera parallel to the monitor screen, each camera was positioned in such a way that the geometric distortions of the obtained reproduction of the test image were symmetric. The test image was displayed on a monitor serving as the panel illumination (Asus PA248QV), and the acquired images were observed in real time on the second monitor. For the time of acquiring the images, the aforementioned plexiglass panel was inserted between the monitor and the used camera.

In order to reduce the noise which is present in the captured images, in the experiment, the ${\bar{I}}_{V}$ image was used, obtained as the average of 100 originally captured images $I_{V_{e}}$. Since the aim of the experiment is evaluation of availability of fitting of the compared vignetting models to the real vignetting, there is no need for calculating of vignetting estimate for each color channel, i.e., $\bar{R}$, $\bar{G}$ and $\bar{B}$, of the ${\bar{I}}_{V}$ image. Therefore, in the experiment, for each camera, as the input image,
(29)$$I_{V}=0.2989\bar{R}+0.5870\bar{G}+0.1140\bar{B}$$
was used. The exposure parameters for each camera were carried out automatically for the first image $I_{V_{1}}$, and, for the rest of the images, i.e., $I_{V_{2}},\ldots ,I_{V_{100}}$, the same parameters were used.

The $I_{V_{e}}$ images were acquired in the darkroom so that additional lights did not interfere with this process. Additionally, in the case of Cam-A and Cam-B, all signaling diodes with which the cameras are equipped were covered. Unfortunately, due to its design (the signaling diode has the form of a ring around the lens), such an operation was impossible with the Cam-C. However, failure to perform this operation had no significant impact on the acquired images $I_{V_{e}}$ due to the relatively high brightness of the imaged panel surface compared to the Cam-C signaling diodes, and quite large distance between the camera and the plexiglass panel.

It is important to notice that, because of the aim of the experiment eventually small errors in, e.g., positioning of the monitor, the plexiglass panel or the cameras, as well as a small non-uniformity of screen illumination, do not influence on qualitative evaluation of the experiment results. Of course, such errors affect the quantitative results of the estimation; however, these errors do not change the judgment of the ability of the tested vignetting models on finding the best approximation $\tilde{V}$ based on the input image $I_{V}$; it is exactly this feature of the vignetting models that is evaluated in the experiment.

In the experiment, four vignetting models were compared, i.e., polynomial 2D (P2D), radial polynomial (RP), smooth local polynomial (SLP), and smooth non-iterative local polynomial (SNILP). The entire experiment (i.e., from image acquisition, through all calculations, up to data presentation) was performed using MATLAB R2021a software package with appropriate Toolboxes, i.e., Image Acquisition, Image Processing, and Curve Fitting.

For implementation of the P2D model, the MATLAB fit function from Curve Fitting Toolbox, which calculates, among others, 2D polynomial approximation, can be used. However, due to the limitation of this function, it can only be used for approximation of polynomials with a maximum degree s=5; in the experiment, the own function, which directly uses ordinary least squares equation for 2D polynomial approximation, was written and used. It is important to notice that the own function, in general, has no limit for degree *s* of the approximation polynomial. Moreover, the own function returns the approximation results faster than the fit function, and the differences between approximation results obtained from the own function and the fit function for data from the experiment are no greater than $10^{-10}$; therefore, from a practical point of view, they are negligible.

The implementation of the RP model uses MATLAB polyfit and polyval functions, which calculate parameters of approximation polynomial and values of approximation function using the obtained parameters. The implemented own function, which carries out these tasks, was no better than the aforementioned MATLAB functions. Additionally, in the case of the RP model, there is a need for founding of optical center *C* of the lens-camera system based on $I_{V}$ image; from this point, the distance *r* is calculated (see Section 1.2). In the experiment, *C* was found by searching of the coordinates of the maximum of 2D polynomial approximation of degree s=2 of the $I_{V}$ image. This approach was compared with others, e.g., a 2D polynomial approximation with degree s=4 and the usage of smoothing function with moving window; however, none of the tested approaches give better results in the sense of values of $\mathrm{STD}\left({\tilde{I}}_{flat}\right)$ and $\mathrm{IQR}\left({\tilde{I}}_{flat}\right)$ than the usage of a 2D polynomial approximation with s=2.

In the case of SLP and SNILP, both models were implemented using polyfit and polyval functions for 1D approximation and calculation approximated values. The results for the SLP model are obtained for k=25 iterations.

All models were compared for different degrees *s* of the approximation polynomials, i.e., for s∈{2,…,10}. The tested range of the values of polynomials degree fully covers the value of the approximation polynomial degree used in practice; in the literature, it is recommended to use *s* from 2, for tasks that do not require precise correction, and up to 6 in more demanding applications.

### 4.2. Results of the Experiment

In Figure 7, acquired calibration images (Figure 7a–c) and $I_{V}$ images (Figure 7d–f) obtained from the tested cameras are presented. The numerical results of the experiment are presented in Table 3 and Table 4. Additionally, in Figure 8, comparison of normalized $I_{V}$ images and the obtained $\tilde{V}$ for s=10 is presented. The normalized images $\mathrm{norm}(I_{V})$ are calculated as follows:(30)$$\mathrm{norm}\;\left(I_{V}\left(i,j\right)\right)=I_{V}\left(i,j\right)\;\cdot \;\max_{\forall i,j}\;{\left(\mathrm{smooth}\;\left(I_{V}\left(i,j\right)\right)\right)}^{-1},$$
where $\mathrm{smooth}\;\left(I_{V}\left(i,j\right)\right)$ is a result of smoothing of the $I_{V}$ image with 2D Gaussian filter with standard deviation σ=0.5 and size 10×10; the $\mathrm{norm}(I_{V})$ images are used only for visualization (these images are not used during calculation of $\tilde{V}$). Figure 9 shows the results of corrections of $I_{V}$ images, i.e., the ${\tilde{I}}_{flat}$ images, obtained for s=10.

### 4.3. Discussion of the Results

The experiment on real data confirms the results of numerical tests (Section 3.3), i.e., for a large number *k* of iterations of the SLP model, the results obtained from the SLP and SNILP models are the same. However, it is important to remember that the computation time which is needed for obtaining the same results from both models is much longer in the case of the SLP model than in the case of the SNILP model.

Comparing results from all tested models, it can be easily seen that, beyond the results for the IQR measure obtained from Cam-B for s∈{2,3}, the order of the obtained results (regardless of the used measure) is the same, i.e., the best results are given by the SLP and SNILP models, slightly worse results are given by the P2D model, and much worse results are obtained from the RP model. It is worth noting that, e.g., for the STD measure and s≥4, the results from the SNILP and SLP models are usually at least 2 times better than the results obtained from the RP model, which is preferred by many authors.

The STD and IQR measures show aggregated information about models quality. Additional knowledge about the models is provided by the graphs presented in Figure 9. Comparing the results for the SPL and SNILP models with the results obtained for the P2D and RP models, it can be seen that the spatial uniformity of the dispersion of values in the ${\tilde{I}}_{flat}$ images are different. For the individual ${\tilde{I}}_{flat}$ images obtained from the SLP and SNILP models, the dispersion of its values is spatially uniform, which indicates—assuming that the mean value of cameras noise $\eta_{Cam}=0$—a good fit of the $\tilde{V}$ estimates to the input images $I_{V}$. Following this line of thinking, assuming that the $I_{flat}$ images were taken correctly, it means that the $\tilde{V}$ estimates obtained from these models exactly approximate the real vignetting *V* of the individual cameras. Contrary to these results, in the ${\tilde{I}}_{flat}$ images obtained from the P2D and RP models, regions exist in which the spatial uniformity is not preserved. This is especially true for the results obtained for Cam-B (Figure 9b,e) and Cam-C (Figure 9c,f). Based on above the assumptions, the occurrence of such areas in the ${\tilde{I}}_{flat}$ image proves that the obtained vignetting estimate $\tilde{V}$ is inaccurately adjusted to the real vignetting *V* of the tested camera.

## 5. Comparison of Others Features of the Tested Vignetting Models

The quality of the vignetting correction results is not the only criterion which should be taken into consideration while choosing a model of vignetting for a specific application. The others are, e.g., computational complexity (determining the speed of calculation), a required memory size (determining the size of images for which the selected model can be used in the given conditions, e.g., using intelligent camera with limited hardware resources), and, last but not least, the minimum amount of data required to save the determined vignetting estimate.

In Table 5, results of the measurement of computation times of the compared vignetting models are presented. Additionally, in Figure 10, the obtained results are presented in the form of a log-log graph. The presented results are obtained as a median value from 10 measures for each tested image format and the vignetting model. For all vignetting models, the degree s=5 of their approximation polynomials was chosen. In the case of the RP model, an image optical center was found using the same method as in the experiment (Section 4.1). For the SLP model, the number of iterations k=10 was chosen. The test vignetting images were obtained using the method presented in Appendix A. The measurements were conducted on a computer equipped with AMD Ryzen 7 3800X processor, 16 GB RAM, and a M.2 PCIe NVMe SSD disk. The results show that the SNILP model, already, for images with a resolution equal to or greater than 1600×1200, is the fastest model. The difference in computation times between the SNILP model and the next model, i.e., RP, grows with increasing image resolution. It is worth noticing that the results obtained for the SLP and SNILP models confirm relation (Equation 26), i.e., that the calculation for the SNILP model is at least *k* times faster than in the case of the usage the SLP model.

The lower bound of a model required memory size can be estimated from the analysis of the minimal memory size needed for calculation of vignetting estimate. In the case of the tested models, for this purpose, Ordinary Least Squares method is used; hence, based on its matrix formulation, i.e.,
(31)$$\tilde{\beta }={\left(X^{T}X\right)}^{-1}X^{T}y,$$
where $\tilde{\beta }$ is a vector of estimated polynomial parameter, X is a matrix of independent variables, and y is a vector of observations; it can be seen that the size of the matrix X, in practice, is determining the amount of needed memory. Let I denote the size of the analyzed image of resolution M×N, where M≤N, and *s* is an order of the used approximation polynomial. The comparison of X size is presented in Table 6; in the table, additional remarks about memory size needed by each vignetting model are also presented. Based on this information, it can be concluded that the SNILP model has the lowest memory requirements among the analyzed models.

The determined vignetting estimate can be saved in form, e.g., a matrix $\tilde{V}$ of size I or a set of parameters. The exact number of needed parameters for each model is presented in Table 7. This comparison shows that the SLP model requires the largest number of parameters. The SNILP model requires N(s+1) parameters less, but, when compared with the P2D model, and especially with the RP model, the number of parameters which are required to save the vignetting estimate $\tilde{V}$ is still large. On the other hand, in order to use the vignetting estimate $\tilde{V}$ saved in the form of a set of parameters, it is necessary to carry out appropriate calculations which require appropriate hardware resources (including memory larger than I needed for save $\tilde{V}$ in form of a matrix) and time. Moreover, in applications where the real-time vignetting correction is needed, e.g., on-line monitoring or quality control, storing $\tilde{V}$ in the form of a matrix is the best solution. In such cases, the advantages of usage a smaller number of parameters, as in the case of P2D and RP models, are no longer relevant.

## 6. Conclusions

Both, i.e., numerical and experimental, results prove that the proposed SNILP vignetting model outperforms the SLP model in both quality of vignetting estimation and the computation speed. The comparison of results obtained from the SNILP model with these obtained from the P2D and RP models shows that the proposed model provides a better vignetting correction, in a sense of the used measures $\mathrm{STD}\left({\tilde{I}}_{flat}\right)$ and $\mathrm{IQR}\left({\tilde{I}}_{flat}\right)$, than both models known from the literature.

Summarizing the results presented in this article, it can be stated that the SNILP vignetting model is a universal vignetting model (similar to the P2D and SLP models). However, compared to these models, the SNILP model is characterized by greater accuracy of the obtained result $\tilde{V}$. It should also be noted that, compared to the widely used RP model, the SNILP model gives for s≥2 better, and, for s≥4, much better, results and can be used for vignetting correction of virtually any lens-camera system (where the usage of the RP model is limited due to the use of the radial function). Moreover, the results from the SNILP model for larger images are obtained in the lowest computation time and the more limited hardware resources can be used for its computing. This last feature is especially important for visual systems that have limited computational capabilities, such as, e.g., smart cameras and sensors, embedded vision systems or vision systems used in unmanned vehicles.

So far, the vignetting models presented in the literature can be divided into two groups, i.e., universal models (e.g., the P2D model), in which usage is not limited to a certain type of vignetting, but their use requires relatively high computing power and dedicated models (e.g., the RP model), which have a relatively low requirement for computing power; however, their usage is limited. Hence, there is no model that would combine its universal application with low computational complexity. Based on the results presented in the article, it can be concluded that the proposed SNILP model combines these requirements and, in this sense, is a novel contribution to solving the vignetting correction issue.

## Figures and Tables

**Figure 1 sensors-21-07086-f001:**
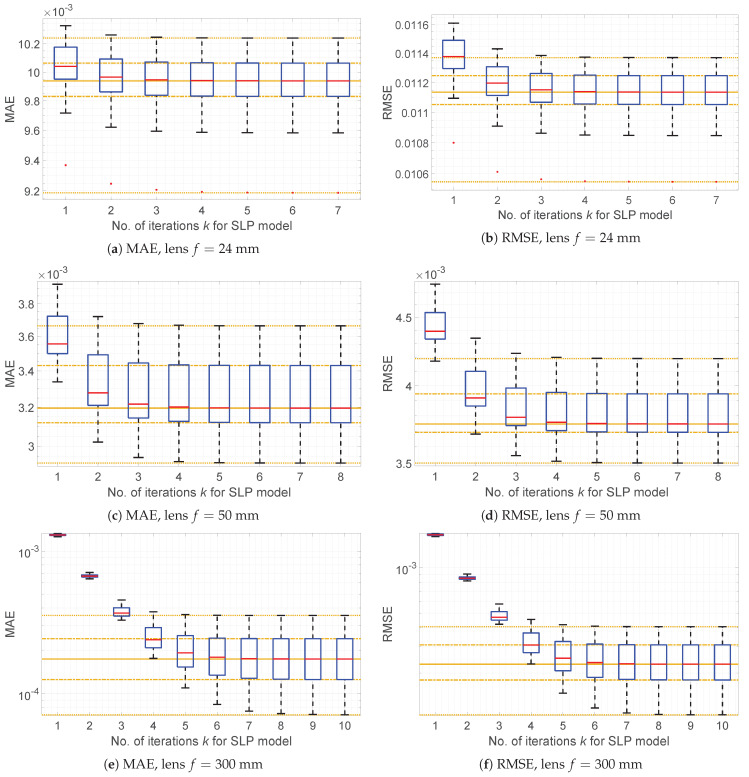
Comparison of MAE and RMSE error values obtained for the LP, SLP, and SNILP models and simulated vignetting of lenses with different focal length *f*; results for *I_V_* contained with Gaussian mixture noise (A5) for *σ_add_* = 5% and *σ_multi_* = 10%. Results for the LP model are equal to results from the SLP model for *k* = 1. Results for the SNILP model are presented as orange lines, whereas lines, from bottom to top, represent minimal, first quartile, median, third quartile, and maximal of error values.

**Figure 2 sensors-21-07086-f002:**
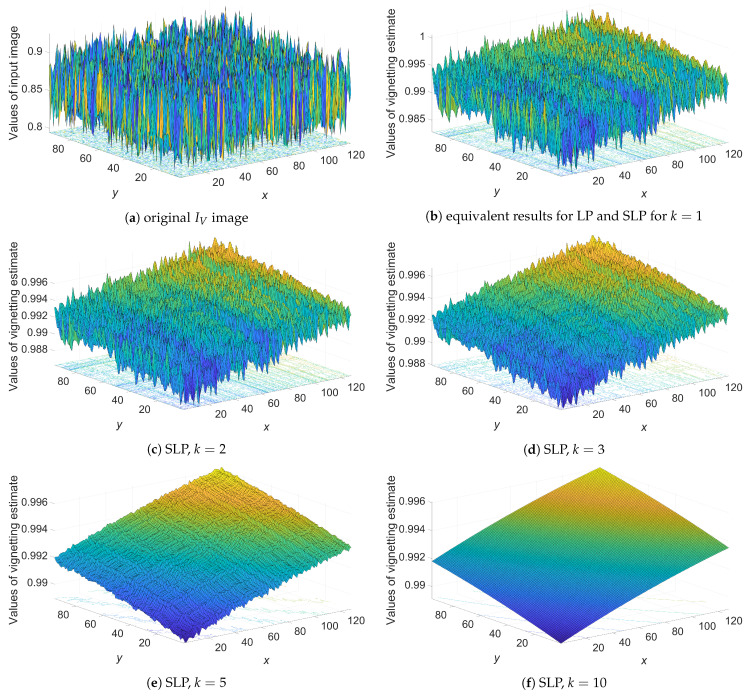
Comparison of surface smoothness of vignetting estimators $\tilde{V}$ obtained from: (**b**) LP and (**b**–**f**) SLP models, (**a**) image *I_V_* of size 640 × 480 pixels contained with Gaussian mixture noise (A5) for *σ_add_* = 5% and *σ_multi_* = 10%.

**Figure 3 sensors-21-07086-f003:**
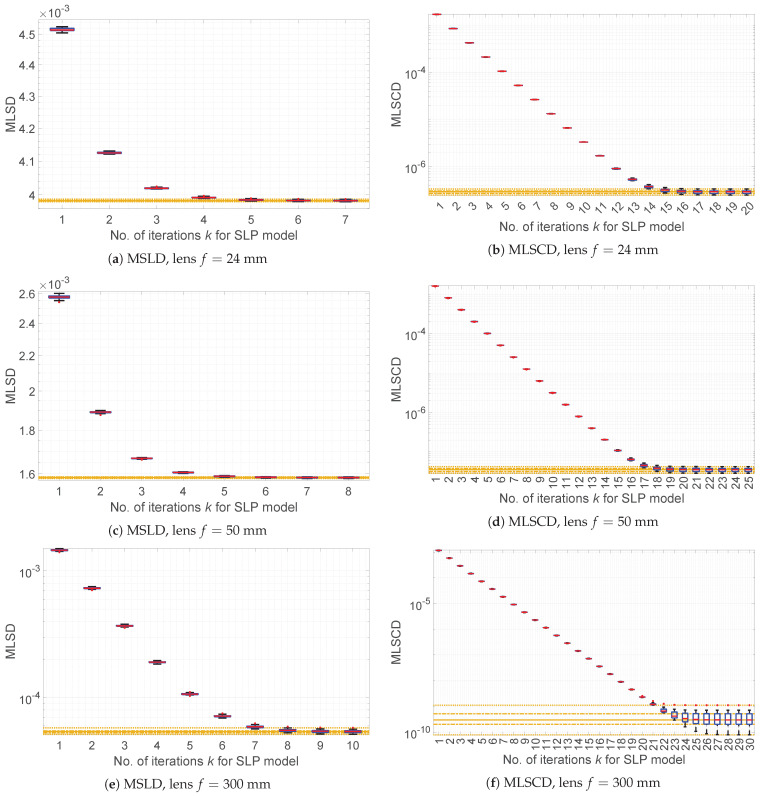
Comparison of MLSD and MLSCD values obtained for the LP, SLP, and SNILP models and simulated vignetting of lenses with different focal length *f*; results for *I_V_* contained with Gaussian mixture noise (A5) for *σ_add_* = 5% and *σ_multi_* = 10%. Results for the LP model are equal to results from the SLP model for *k* = 1. Results for the SNILP model are presented as orange lines, whereas lines, from bottom to top, represent minimal, first quartile, median, third quartile, and maximal of error values.

**Figure 4 sensors-21-07086-f004:**
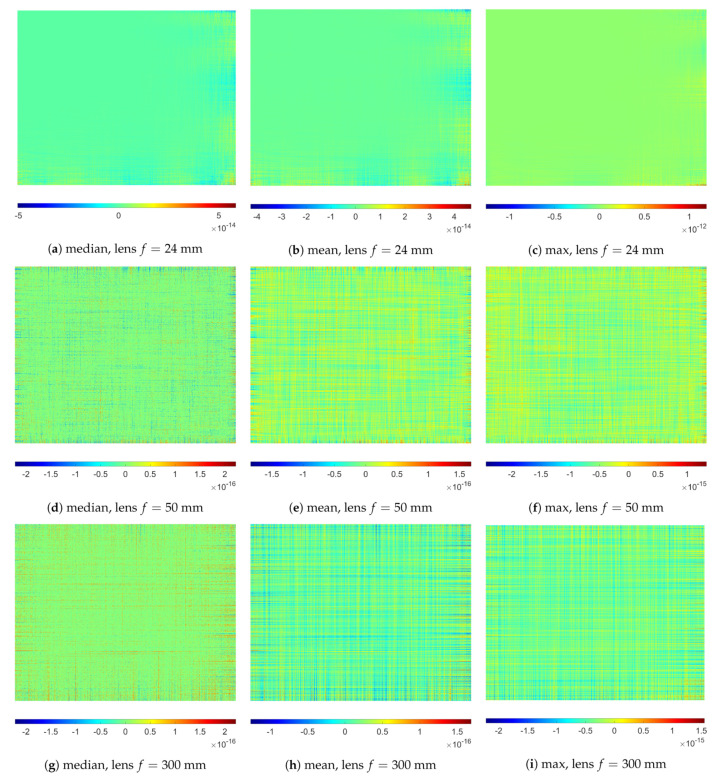
Errors related to using SNILP model with different sequence of approximation directions, examples for images *I_V_* of resolution 1280 × 1024 pixels contained with Gaussian mixture noise (A5) for *σ_add_* = 5% and *σ_multi_* = 10%.

**Figure 5 sensors-21-07086-f005:**
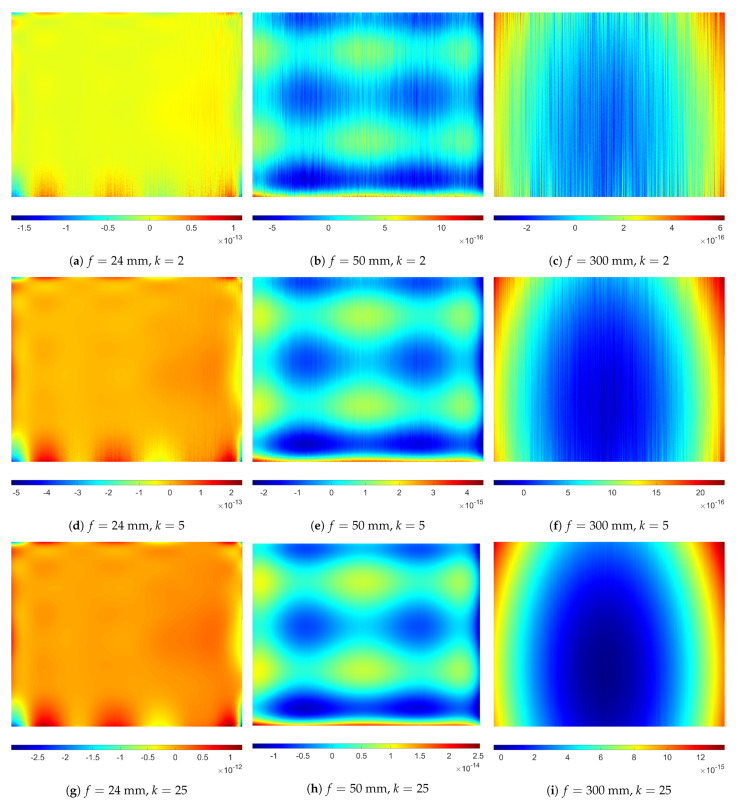
Mean errors $\bar{\varepsilon }\left(i,j\right)$ related to using SNILP in an iterative manner for different numbers of repetitions *k*, results for image *I_V_* of size 1280 × 1024 pixels contained with Gaussian mixture noise (A5) for *σ_add_* = 5% and *σ_multi_* = 10%.

**Figure 6 sensors-21-07086-f006:**
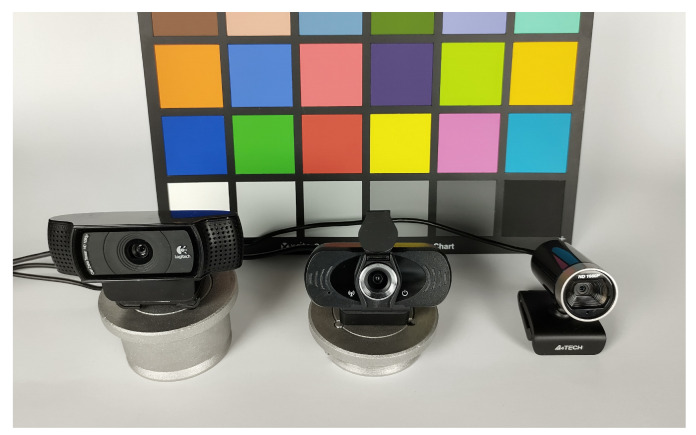
Cameras used in the experiment, from left to right: Logitech C920 (Cam-A), Xiaomi IMILAB CMSXJ22A (Cam-B), and A4Tech PK-910H (Cam-C).

**Figure 7 sensors-21-07086-f007:**
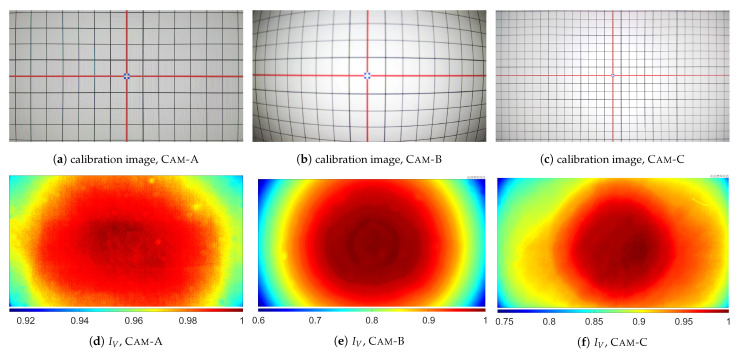
Comparison of acquired calibration images and *I_V_* images obtained from the tested cameras.

**Figure 8 sensors-21-07086-f008:**
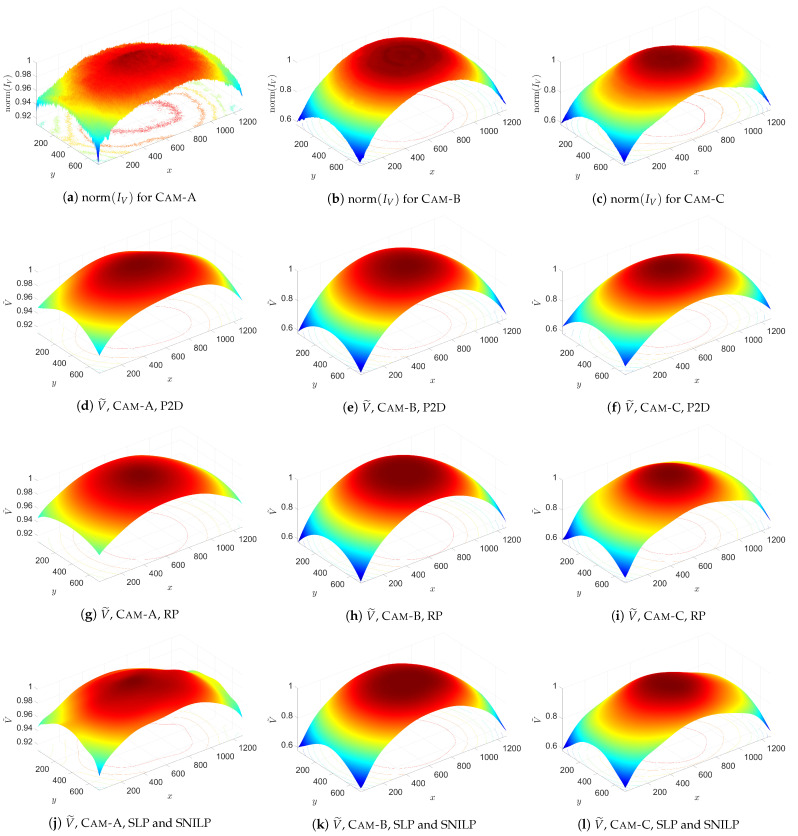
Comparison of vignetting estimates $\tilde{V}$ obtained for the tested cameras and vignetting models for degree *s* = 10 of the used approximation polynomials. The charts (**j**–**l**) present results obtained for the SLP and SNILP models, which, in the case of the used parameter *k* = 25 (number of iterations), in the SPL model, are equivalent.

**Figure 9 sensors-21-07086-f009:**
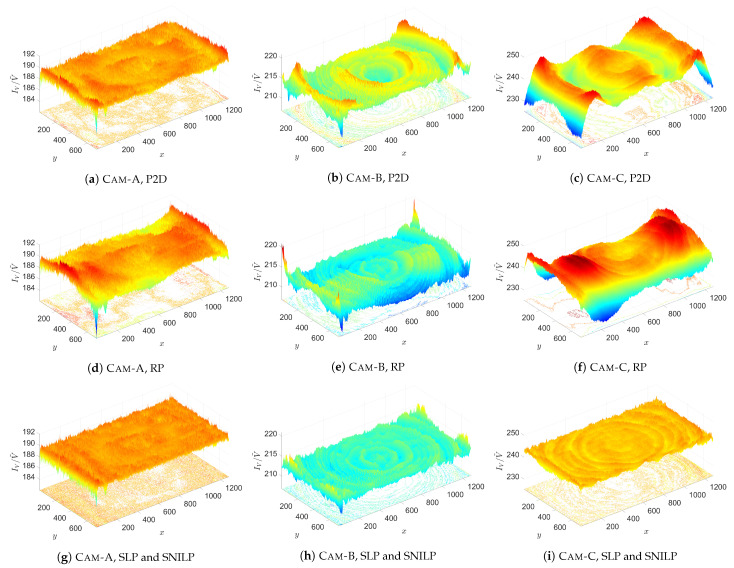
Comparison of vignetting correction of flat-field image *I_V_* for the tested cameras and vignetting models for degree *s* = 10 of the used approximation polynomials. The charts (**j**–**i**) present results obtained for the SLP and SNILP models, which, in the case of the used parameter *k* = 25 (number of iterations), in the SPL model, are equivalent.

**Figure 10 sensors-21-07086-f010:**
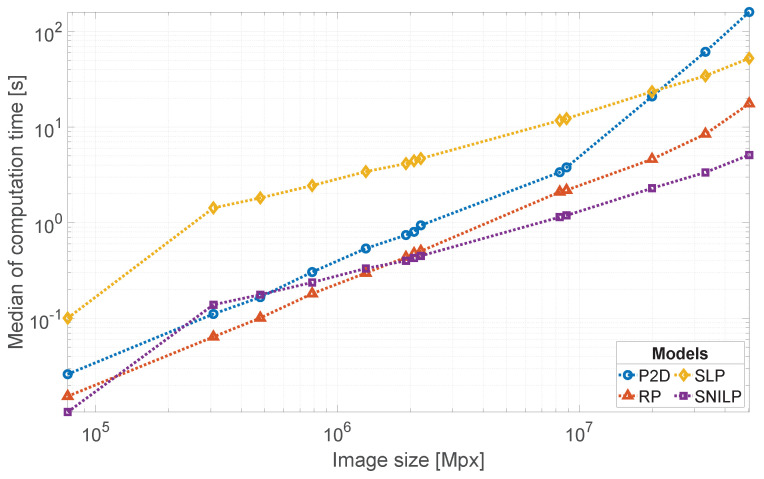
Comparison of computation times of the tested models for selected image formats and the degree of approximation polynomial s=5. Exact information about chosen image formats is given in Table 5.

**Table 1 sensors-21-07086-t001:** Differences in values of vignetting estimators for using the SNILP model with different sequences of approximation direction.

Percentile	Focal Length [mm]		
	24	50	300
0	0.0	0.0	0.0
0.05	1.11e-16	0.0	0.0
0.25	4.44e-16	1.11e-16	1.11e-16
0.5	1.67e-15	1.11e-16	1.11e-16
0.75	7.44e-15	2.22e-16	3.33e-16
0.95	3.26e-14	4.44e-16	4.44e-16
1	7.04e-13	1.89e-15	1.55e-15
mean	7.13e-15	1.80e-16	1.83e-16

**Table 2 sensors-21-07086-t002:** Main technical data of the cameras used in the experiment.

Parameters	Webcam		
	Cam-A	Cam-B	Cam-C
	Logitech C920	Xiaomi IMILAB CMSXJ22A	A4Tech PK-910H
Diagonal field of view	78°	85°	70°
Maximal resolution	1920×1080	1920×1080	1920×1080
Maximal frame rate @ 1080p	30 fps	30 fps	30 fps
Focus type	auto focus	fixed focus	fixed focus
Focus range	—	—	>60 cm

**Remarks**: All data are provided by camera manufacturers.

**Table 3 sensors-21-07086-t003:** Comparison of $\mathrm{STD}\left({\tilde{I}}_{flat}\right)$ values.

Camera	Model	Order								
		2	3	4	5	6	7	8	9	10
**Cam-A**	P2D	0.4831	0.4802	0.3932	0.3371	0.2930	0.2837	0.2746	0.2677	0.2577
	RP	0.6181	0.6181	0.5858	0.5858	0.5837	0.5837	0.5828	0.5828	0.5805
	SLP	0.4485	0.4338	0.3254	0.3148	0.2796	0.2732	0.2520	0.2493	0.2458
	SNILP	0.4485	0.4338	0.3254	0.3148	0.2796	0.2732	0.2520	0.2493	0.2458
**Cam-B**	P2D	2.7785	2.7503	0.9788	0.9395	0.9299	0.9224	0.6145	0.6017	0.5447
	RP	2.8143	2.8143	1.2296	1.2296	1.2322	1.2322	1.0423	1.0423	1.0143
	SLP	2.2618	2.2276	0.9267	0.9222	0.6818	0.6665	0.5295	0.5178	0.4965
	SNILP	2.2618	2.2276	0.9267	0.9222	0.6818	0.6665	0.5295	0.5178	0.4965
**Cam-C**	P2D	2.0798	1.8989	1.7279	1.6951	0.8810	0.8221	0.7058	0.6857	0.5785
	RP	2.8762	2.8762	2.6539	2.6539	2.2816	2.2816	2.2510	2.2510	2.2501
	SLP	2.0720	1.8825	1.5142	1.4789	0.6578	0.6446	0.5635	0.5408	0.4378
	SNILP	2.0720	1.8825	1.5142	1.4789	0.6578	0.6446	0.5635	0.5408	0.4378

**Remarks**: For camera cells, background colors (from blue, through light blue, light red to red) represent range (from the best to the worst one) of the model results based on the $\mathrm{STD}\left({\tilde{I}}_{flat}\right)$ values.

**Table 4 sensors-21-07086-t004:** Comparison of $\mathrm{IQR}\left({\tilde{I}}_{flat}\right)$ values.

Camera	Model	Order								
		2	3	4	5	6	7	8	9	10
**Cam-A**	P2D	0.6565	0.6629	0.5102	0.4101	0.3563	0.3512	0.3363	0.3302	0.3088
	RP	0.8365	0.8365	0.7544	0.7544	0.7459	0.7459	0.7415	0.7415	0.7421
	SLP	0.6156	0.5762	0.3974	0.3818	0.3480	0.3408	0.2989	0.2951	0.2901
	SNILP	0.6156	0.5762	0.3974	0.3818	0.3480	0.3408	0.2989	0.2951	0.2901
**Cam-B**	P2D	2.7444	2.6071	1.2456	1.1831	1.0988	1.0824	0.6802	0.6854	0.6690
	RP	2.8675	2.8675	1.6103	1.6103	1.6080	1.6080	1.3072	1.3072	1.3121
	SLP	3.2378	3.2712	1.1637	1.1685	0.8412	0.8102	0.6674	0.6548	0.6277
	SNILP	3.2378	3.2712	1.1637	1.1685	0.8412	0.8102	0.6674	0.6548	0.6277
**Cam-C**	P2D	2.9902	2.9323	2.2331	2.2332	1.0541	1.0078	0.9222	0.8944	0.7722
	RP	4.0800	4.0800	3.1657	3.1657	2.7657	2.7657	2.6463	2.6463	2.6344
	SLP	2.8820	2.8441	2.1795	2.1818	0.8612	0.8455	0.7420	0.7182	0.5849
	SNILP	2.8820	2.8441	2.1795	2.1818	0.8612	0.8455	0.7420	0.7182	0.5849

**Remarks**: For camera cells, background colors (from blue, through light blue, light red to red) represent range (from the best to the worst one) of the model results based on the $\mathrm{IQR}\left({\tilde{I}}_{flat}\right)$ values.

**Table 5 sensors-21-07086-t005:** Computation times of the tested models for selected image formats and the degree of approximation polynomial s=5.

Image Resolution	Image Size [Mpx]	Image Format	Computional Time [s] for Model			
			P2D	RP	SLP	SNILP
320×240	0.08	QVGA	0.03	0.02	0.10	0.01
640×480	0.31	VGA	0.11	0.06	1.43	0.14
800×600	0.48	Super VGA	0.17	0.10	1.81	0.18
1024×768	0.79	XGA	0.30	0.18	2.44	0.24
1280×1024	1.31	SXGA	0.54	0.30	3.42	0.33
1600×1200	1.92	UXGA	0.74	0.44	4.14	0.40
1920×1080	2.07	HDV	0.80	0.48	4.43	0.43
2048×1080	2.21	2K Digital Cinema	0.94	0.50	4.68	0.45
3840×2160	8.29	4K UHDTV	3.36	2.10	11.70	1.14
4096×2160	8.85	Canon PowerShot G9 X Mark II	3.78	2.18	12.22	1.19
5472×3648	19.96	4K Digital Cinema	20.85	4.60	23.40	2.29
7680×4320	33.18	8K UHDTV	61.03	8.43	34.21	3.35
8688×5792	50.32	Canon EOS 5DS	158.97	17.48	52.16	5.10

**Remarks**: For each image resolution, color of cell background (from blue, through light blue, light red to red) represents range (from the best to the worst one) of the vignetting model computational times.

**Table 6 sensors-21-07086-t006:** Comparison of models required memory size.

Model	Size of *X*	Remarks
P2D	$\frac{1}{2}\left(s^{2}+3s+2\right)I$	
RP	(s+1)I	In the case of this model, the stage of searching for an optical center of the $I_{V}$ image can require more memory than the stage of approximation of the radial function $\tilde{V}$. For the approach to searching for the optical center, which is used in the experiment above, the statement is true when $$\frac{1}{2}\left(s_{c}^{2}+3s_{c}+2\right)I>\left(s+1\right)I,$$ where $s_{c}$ is the order of 2D polynomial used for image optical center searching.
SLP	$\frac{1}{M}\left(s+1\right)I$	In the SPL model, there is a stage of calculation vignetting estimate $\tilde{V}$ by averaging (Equation 4) the previously calculated values ${\tilde{V}}_{x}$ and ${\tilde{V}}_{y}$, which requires additional space of a size 2I to keep these variables. Because of the small value of *s*, which is commonly used, the stage of averaging usually requires more memory than 1D approximation needed for calculation of ${\tilde{V}}_{x}$ and ${\tilde{V}}_{y}$.
SNILP	$\frac{1}{M}\left(s+1\right)I$	

**Table 7 sensors-21-07086-t007:** The number of parameters required for saving vignetting estimation results.

Model	Type of Parameters	Number of Parameters
P2D	parameters of 2D approximation polynomial	$\frac{1}{2}\left(s^{2}+3s+2\right)$
RP	parameters of 1D radial polynomial function + coordinates of image optical center	s+3
SLP	parameters of 1D approximation of each line in the horizontal and the vertical direction of the input image $I_{V}$	(M+N)(s+1)
SNILP	parameters of 1D approximation of each line along the larger side of the input image $I_{V}$	M(s+1), where M≤N

## Data Availability

The data presented in this study are available on reasonable request from the corresponding author.

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
