# Peer review of "A Smooth Non-Iterative Local Polynomial (SNILP) Model of Image Vignetting"

_sensors, 2021, doi:10.3390/s21217086_

Round 1

Reviewer 1 Report

The article provides an interesting background to image vignetting and presents a method that achieves a state-of-the-art level of IV correction with comparably less computational complexity. The structure of the article is clear and it is well-written, but the authors could clarify a minor point:

In figures 8 and 9, the last 3 graphs show results for "SLP and SNILP" which can lead to the confusion that the results are for the algorithms applied together rather than that the results are equivalent for each which was noted further away in the text. Could clarify close to the figure.

Author Response

Reviewer 1: The article provides an interesting background to image vignetting and presents a method that achieves a state-of-the-art level of IV correction with comparably less computational complexity. 

Authors: We appreciate very much the positive feedback from the Reviewer.

Reviewer 1: The structure of the article is clear and it is well-written, but the authors could clarify a minor point:

In figures 8 and 9, the last 3 graphs show results for "SLP and SNILP" which can lead to the confusion that the results are for the algorithms applied together rather than that the results are equivalent for each which was noted further away in the text. Could clarify close to the figure.

Authors: Thank you for this valuable comment. In response, the captions under Figures 8 and 9 have been supplemented with information expressly describing the fact that the graphs mentioned by Reviewer show equivalent results obtained independently for the SLP (for iterations number k = 25) and SNILP models. 

Additionally, the article was searched for similarly ambiguous descriptions and, correcting the caption under Figure 2 (a).

Reviewer 2 Report

This paper presents a new model named as Smooth Non-Iterative Local Polynomial (SNILP) for image vignetting. And experimental results demonstrated the effectiveness of the proposed technique. A few detailed comments are listed as follows:

  1. The contribution and innovation in this manuscript are ambiguous no matter in the introduction or from the perspective of the exhibition in the main body. It would be great to concise your contributions from the technique view and highlight your innovation.
  2. Some of the equations and figures are not explicitly illustrated. Please thoroughly check the paper. For example, the meaning of PolyReg_x^s and others should be concisely illustrated when the first time appeared. And I am also confused about the Fig.1, where the title shows ‘for LP, SLP and SNLP’ while the figures shown seemingly are for SLP.
  3. The readability should be improved overall. The paper writing of whole manuscript should be required and polished to avoid ambiguity and obscurity of expression or colloquial writing. Please carefully handle sentences like: ``… the values of approximations PolyReg^s _x and PolyReg^s _v, respectively, for horizontal and vertical lines of IV are calculated in sequences, e.g., as a first the approximation in horizontal direction is calculated, and than, based on this results, the in approximation in vertical direction is calculated”.

Author Response

Reviewer 2: This paper presents a new model named as Smooth Non-Iterative Local Polynomial (SNILP) for image vignetting. And experimental results demonstrated the effectiveness of the proposed technique. A few detailed comments are listed as follows:

Authors: Thank you very much for your valuable comments.

Reviewer 2:
1. The contribution and innovation in this manuscript are ambiguous no matter in the introduction or from the perspective of the exhibition in the main body. It would be great to concise your contributions from the technique view and highlight your innovation.

Authors: In response to this comment, Chapter 6 was supplemented with a brief recapitulation of the meaning of the developed SNILP model for the vignetting correction problem.

Reviewer 2:
2. Some of the equations and figures are not explicitly illustrated. Please thoroughly check the paper. For example, the meaning of PolyReg_x^s and others should be concisely illustrated when the first time appeared. And I am also confused about the Fig.1, where the title shows ‘for LP, SLP and SNLP’ while the figures shown seemingly are for SLP.

Authors: The article has been carefully searched for ambiguous descriptions. As a result, e.g., descriptions used in Figures 1-3 have been corrected, sections 2.1, 2.2 and 3.1 have been redrafted, and descriptions accompanying some formulas (e.g. (26) and (31)) have also been corrected.

Reviewer 2:
3. The readability should be improved overall. The paper writing of whole manuscript should be required and polished to avoid ambiguity and obscurity of expression or colloquial writing. Please carefully handle sentences like: ``… the values of approximations PolyReg^s _x and PolyReg^s _v, respectively, for horizontal and vertical lines of IV are calculated in sequences, e.g., as a first the approximation in horizontal direction is calculated, and than, based on this results, the in approximation in vertical direction is calculated”.

Authors: In response to this remark, we carefully inspected the article. Over 100 errors of different types have been found and corrected. We hope that the introduced changes have improved the readability of the article.

Round 2

Reviewer 1 Report

I am satisfied with the corrections made.